# Geospatial and phylogenetic clustering of acute and recent HIV infections in Lilongwe, Malawi

Griffin J. Bell[1,2]*, Kimberly A. Powers[1], Oliver Ratmann[3], Ann M. Dennis[4], Pearson Mmodzi[5], Mitch Matoga[5], Edward Jere[5], Jane S. Chen[6], Courtney N. Maierhofer[1], Sarah E. Rutstein[4], Kathryn E. Lancaster[7], Maganizo B. Chagomerana[4,5], Naomi Bonongwe[5], Esther Mathiya[5], Beatrice Ndalama[5], David Bonsall[8], Sharon S. Weir[1], Mina C. Hosseinipour[4,5], Michael Emch[1,9], Myron S. Cohen[1,4,10], Irving F. Hoffman[4,5], William C. Miller[1]

1 Department of Epidemiology, Gillings School of Global Public Health, University of North Carolina, Chapel Hill, North Carolina, United States of America, 2 Department of Epidemiology, Bloomberg School of Public Health, Johns Hopkins University, Baltimore, Maryland, United States of America, 3 Department of Mathematics, Imperial College London, London, United Kingdom, 4 Division of Infectious Diseases, School of Medicine, University of North Carolina, Chapel Hill, North Carolina, United States of America, 5 University of North Carolina Project, Lilongwe, Malawi, 6 Department of Health Behavior, Gillings School of Global Public Health, University of North Carolina, Chapel Hill, North Carolina, United States of America, 7 Department of Implementation Science, School of Medicine, Wake Forest University, Winston-Salem, North Carolina, United States of America, 8 Nuffield Department of Medicine, University of Oxford, Oxford, United Kingdom, 9 Department of Geography, University of North Carolina, Chapel Hill, North Carolina, United States of America, 10 Department of Microbiology and Immunology, School of Medicine, University of North Carolina, Chapel Hill, North Carolina, United States of America

* gbell19@jhu.edu

## Abstract

HIV transmission during early HIV infection impedes efforts to end HIV as a public health threat, as diagnosis typically occurs after this period of elevated transmission risk. To guide diagnosis and prevention strategies, we evaluated the geospatial and phylogenetic clustering of acute and recent HIV infection in Lilongwe, Malawi. We identified people with acute (pre-seroconversion) HIV infection (AHI) and a random sample of people with post-acute HIV infection who presented to a sexually transmitted infections (STI) clinic in Lilongwe, Malawi between 2015 and 2019. We evaluated infection recency in people with post-acute HIV using a LAg-Avidity assay. We mapped the household locations of people with AHI and identified geospatial clusters using a flexible scan statistic. We constructed consensus sequences from deep sequencing reads to identify phylogenetic clusters through genetic distance thresholds and maximum likelihood trees. We identified 141 people with AHI, 30 people with recent HIV, and 652 people with chronic (non-recent) HIV. We identified four geospatial clusters that contained the residences of 30% of clinic attendees with AHI, despite comprising just 0.8% of the populated land area and 3.5% of the population. We also identified fourteen distinct two-person phylogenetic clusters. Ten of the fourteen were male-female pairs, nine of which were clinic referral pairs. The remaining four were same-sex pairs who had not referred each other to the clinic and may have

**Data availability statement:** The data required to replicate study results include household GPS locations which cannot be transferred to a third party without a data use agreement, according to the policies of the University of North Carolina at Chapel Hill. Deidentified data still contain information derived from human participants and are therefore subject to ethical and legal restrictions on public sharing set by the University of North Carolina at Chapel Hill. Requests for complete or de-identified data can be directed to the authors and will be processed under the guidance of the University of North Carolina Office of Human Research Ethics (contact: 919-966-3113). Data from external sources that the authors cannot redistribute, but can be downloaded or requested, include administrative boundaries (gadm.org), the high resolution settlement layer (datafor-good.facebook.com), and the 2018 Malawian Census (nsomalawi.mw). Additional HIV sequences are available from the Los Alamos National Lab database (hiv.lanl.gov).

**Funding:** Funding was provided by United States National Institutes of Health (NIH) through the following grants: R01AI114320 (KAP, WCM), F31AI167672 (GJB), T32AI007001 (GJB). The funders had no role in study design, data collection and analysis, decision to publish, or preparation of the manuscript.

**Competing interests:** The authors have declared that no competing interests exist.

been missing network intermediaries. Three of the fourteen phylogenetic pairs consisted of only acute/recent members, and zero phylogenetic linkages were located within geospatial clusters. AHI detection programs anchored in STI clinic populations and their neighborhoods could facilitate identification of early HIV infection, enabling treatment initiation and transmission prevention efforts during this most infectious period. Future studies of intervention packages and deployment approaches can help inform the optimal design and implementation of AHI-focused strategies for reducing HIV incidence.

## Introduction

HIV transmission during early HIV infection may disproportionately contribute to population-level HIV incidence [1,2]. Elevated viral loads [3,4] and decreased HIV status awareness [5,6] lead to increased transmission rates [7–11] during the first months of infection, especially during acute infection (the first 4–6 weeks of infection, before seroconversion). Rapid propagation of HIV during early infection could hinder the impact of HIV treatment as prevention interventions, since identification and treatment of infection typically occurs well after this phase ends [5]. Despite widespread antiretroviral treatment, HIV incidence declines have not reached UNAIDS targets, suggesting that additional interventions will be needed to achieve HIV elimination [12,13]. Strategies that focus on preventing transmission during early (including acute) HIV infection offer a potentially promising approach [1,2].

Because incident [14–16] and prevalent [15,17–19] HIV infections are heterogeneously distributed across space and demographic groups [12,20–22], interventions concentrating on "key" populations and geographies can be efficient in reducing population-level HIV incidence. Such interventions could avert 10 million incident HIV infections over 15 years in sub-Saharan Africa, compared to 5.3 million achievable with uniformly distributed interventions for the same cost [23]. Acute HIV infection (AHI) may play a larger role in population-level HIV incidence in these populations: modeling studies estimate a larger contribution of transmissions during AHI in populations with greater partner concurrency and switching [9,24].

The identification of geographic areas with high AHI prevalence or evidence of rapid HIV propagation during AHI would suggest that geographically tailored interventions with an AHI focus, such as increased AHI surveillance in neighborhoods with recently detected acute infections, could interrupt prospective HIV transmission chains. Alternatively, if HIV transmission chains are spatially diffuse, but suggestive of a large contribution of transmission during AHI, then AHI-focused interventions propagated along sexual networks may be more efficient than geographically guided interventions. A more limited role of AHI in transmission chains would suggest that interventions without an AHI detection component may be sufficient.

In this analysis, we aimed to inform HIV intervention approaches by leveraging geospatial and viral genetic data collected from people presenting to an STI clinic in Lilongwe, Malawi between 2015 and 2019, including 141 people diagnosed during

AHI. With these data, we sought to: 1) identify geospatial and spatiotemporal clusters of AHI and characterize their proximity to major roads and venues where people meet new sexual partners, 2) identify phylogenetically linked acute and recent infections (potential transmission chains), and 3) evaluate the overlap between spatial and spatiotemporal clusters and phylogenetic links.

## Methods

### Study design and population

We used data from the iKnow study (NCT02467439), which was a clinic-based, randomized, controlled study evaluating the impact of a combination HIV detection intervention on HIV status awareness [25] conducted between 2015 and 2019 in two STI clinics in Lilongwe, Malawi. One clinic, at Kamuzu Central Hospital, closed in 2015, so study participants were predominantly drawn from Bwaila STI Clinic, which is the only free STI clinic in Lilongwe District. The study population consisted of STI clinic patrons (index participants), their sexual partners, and their social contacts. Eligible index participants were STI clinic patients who were diagnosed with acute or seropositive HIV, at least 18 years of age, living in Lilongwe, sexually active in the six months before enrollment, and willing to provide informed consent. Eligible partners and social contacts were those 18 years or older who were referred to the clinic by index participants, willing to provide informed consent, and planning to remain in Lilongwe for the duration of the study. All people with AHI and people with a seropositive HIV infection who were randomized to the active arm of the iKnow study had blood samples collected [25].

### Acute, recent, and chronic HIV infection definitions

Per Malawian standard of care for STI clinic patients, all participants received HIV testing with serial rapid serological tests. AHI was defined as an initial negative rapid test (seronegative) or an initial positive rapid test followed by a negative rapid test (serodiscordant), with the subsequent detection of HIV RNA > 5,000 copies/ml through polymerase chain reaction (PCR). After a positive HIV RNA PCR, the original specimen was reassessed using two rapid antibody tests and/or a standard HIV ELISA to confirm the initial negative/discordant result. Among people with two positive rapid tests (seropositive) and a blood sample, assays to identify recent HIV infection (RHI) (mean time since infection acquisition ~4 months, but not acute) were performed using the Sedia HIV-1 LAg-Avidity EIA (Sedia Biosciences Corporation, Oregon, USA), the CDC LAg-Avidity Assay Data Management File, and the Abbott RealTime HIV-1 m2000 platform (Abbott Laboratories, Germany) [26]. People with seropositive results not identified as having recent infection were classified as having chronic HIV infection (CHI).

### Geospatial and spatiotemporal analyses

Participants who were seronegative or serodiscordant based on the serial rapid testing protocol completed a locator form at enrollment to allow for tracing in the event of a positive HIV RNA PCR. Using these locator forms, we obtained household point locations for each person with AHI and a random sample of 250 of those without HIV selected through date-ordered systematic sampling [27]. Locator forms for seropositive participants were not collected; therefore, people with post-acute infections did not have household location information. We defined the clinic catchment area as the bounding box of all household locations, encompassing the majority of Lilongwe City and surrounding rural areas. In analyses, we excluded areas with a population of 0 (unpopulated).

We evaluated the spatial clustering of AHI by first calculating the proportion of all residents aged 15–64 years in ~500x500 meter squares who attended the STI clinic with AHI during the study period. We chose to use proportions (higher yield of cases per population) rather than counts (total count of cases regardless of the underlying population denominator) because we considered proportions to be better markers of efficiency when tailoring spatial interventions. The population size (i.e., our denominator) in each square was determined using a high-resolution settlement layer from

Center for International Earth Science Information Network (CIESEN) gridded population data [28], aggregated into desired square sizes and adjusted by the proportion of residents in each traditional authority aged 15–64 years according to the 2018 Malawian Census. We identified spatial clusters of proportions of residents who attended the clinic with AHI using Tango and Takahashi's flexible scan statistic ($\alpha = 0.1$), using our ~500x500 meter squares as the spatial analysis unit and implemented with rflexscan [29]. We also conducted a sensitivity analysis with a resolution of ~1x1 km.

We evaluated the spatiotemporal clustering of AHI through a space-time permutation test (with a more generous $\alpha = 0.5$ to consider more potential clusters, given the search across both space and time) using household point locations as the spatial analysis unit and implemented in SaTScan [30]. In short, this method determines whether there are areas and time periods where cases cluster together more tightly than would be expected if the cases were randomly distributed across space and time. We allowed for elliptical clusters to better capture clusters that might appear along roads.

To better understand the spatial distribution of STI clinic attendance, we also explored the spatial and spatiotemporal clustering of the residential locations of HIV-negative attendees and compared them to the analogous AHI clusters. Both types of clusters may represent areas with high rates of sexual partner acquisition and STI risk, and the extent to which they overlap may help to inform the selection of areas for AHI-focused interventions versus more general HIV/STI prevention approaches.

To assess potential correlations between the residential locations of people with AHI and the locations of venues where people meet new sexual partners, we obtained data on venues identified through community informants as places where people meet new partners using the PLACE method in 2016 [31]. During the PLACE study, all venues (e.g., bars, brothels, resthouses, massage parlors) had location information collected at the traditional authority level, which is the second-smallest census-designated level and is nested within the district level. Individual traditional authorities within Lilongwe city are referred to as "areas" (e.g., Area 18). A random sample (~60%) of all venues had GPS point locations collected during the original PLACE study. We completed the collection of point locations for the remaining 40% of venues in 2022, excluding bar-like venues (e.g., bars, restaurants) due to limited resources. To compare the spatial distribution of venues to AHI clusters, we calculated the number and proportion of venues within AHI clusters using venue point locations. We also calculated the number and proportion of venues in the same traditional authorities as those that contained over 50% of the area of any AHI cluster. Road shapefiles were obtained through OpenStreetMap to assess their proximity to, and intersection with, any identified clusters and areas with a large number of venues [32].

## Sequencing and consensus assembly

Deep sequencing was done in collaboration with the PANGEA consortium [33]. Plasma (500ul) was extracted using the Biomerieux Easymag extraction platform and samples were subject to 12 PCR cycles during library preparation. Libraries were pooled by plate and resulting pools were cleaned up and split into high (500kb+) and low molecular weight size fractions. All high molecular weight fractions and 10% of the low molecular weight fraction mass underwent 12 additional PCR cycles using an xGen Hybridization Capture kit and a custom probe panel to pull out HIV-1 subtypes A, B, C, and D. Captured pools were cleaned and pooled for sequencing using an Illumina Novaseq6000. Whole genome consensus sequences were assembled using *shiver* [34].

HIV strain typing was conducted with Genome Detective using consensus sequences for the whole genome [35]. Whole genome type-C sequences were aligned and cut into *gag* and *pol* genes using the Los Alamos National Lab (LANL) genecutter tool. We downloaded all type-C HIV sequences with >90% coverage from Malawi in the LANL database (n = 1296 for *gag,* n = 1318 for *pol*) and reduced these datasets to 25 sequences with minimal loss to diversity for each of *gag* and *pol* using Treemmer [36]. These sequences were added to our alignment using MAFFT [37].

## Phylogenetic analyses

Evolutionary model testing and tree-building were done using IQTree [38]. The best evolutionary model was selected using the Bayesian information criterion. We visualized the resulting maximum likelihood trees using ggtree in R [39].

We defined phylogenetic clusters as monophyletic sequences with genetic distances <2% in *gag* or *pol*. If less than 50% overlap between *gag* and *pol* in a sequence pair was present, we allowed for sequences to cluster with <4.5% distance in the whole genome [40]. These combined region-specific and whole-genome criteria allowed us to detect potential transmission pairs while remaining robust in the presence of recombination and any partial sequencing failures. Higher thresholds when *gag* and *pol* are not present reflect increased mutation rates in other areas of the genome. Genetic distances were calculated using the Tamura-Nei model with pairwise deletion for missing base positions. We described the resulting phylogenetic clusters by summarizing temporal (time between clinic visits), genetic, and (for acute-acute pairs only) geospatial distances for each linkage pair.

## Results

### Geospatial and spatiotemporal analyses

All 250 people without HIV and 135/141 (96%) people with AHI had household geolocations available. AHI cases were predominantly concentrated in specific population-dense areas in the southern and western portions of the city: 70% (94/135) of people with AHI resided inside, or within 250 meters of, Areas 23, 24, 36, 56, or 57 (Fig 1). We identified four geospatial AHI clusters that captured 40/135 (30%) people with AHI who were identified at the Bwaila clinic and had residential geolocations available between 2015 and 2019 (Table 1, Fig 1). These four areas ranged in size from 1.2 to 1.7 $km^2$, encompassing 0.8% of the populated clinic catchment area and the residences of 31,000 people aged 15–64 (3.5% of the population). Our sensitivity analysis using ~1x1 $km^2$ units mostly identified similar areas with elevated proportions of the population attending Bwaila STI clinic with AHI (Fig A in S1 Text): clusters A1, A2, and A4 were moderately expanded while A3 was absent. Clusters identified in this sensitivity analysis were less spatially efficient for acute detection than were those in the main analysis, capturing 36/135 (27%) AHI cases in 1.5% of the populated clinic catchment area and 4.1% of the population. We also identified six geospatial clusters of clinic attendees without HIV; four directly overlapped with or were adjacent to three of the AHI geospatial clusters identified in the main analysis, and two were stand-alone HIV-negative attendee clusters (Fig B in S1 Text).

Point locations were available for 284/318 (89%) non-bar-like venues and 437/746 (59%) bar-like venues. Traditional authority locations were available for 1,025/1,064 (96%) total venues. The four geospatial AHI clusters contained only 36/721 (5.0%) venues with point locations. However, major roads, namely the M1, S124, and Lilongwe Bypass (which connects the M1 and S124) are within 500 meters of all four geospatial clusters (Fig 2). Notably, these roads also directly connect to Areas 58 and 8, which contained 128 venues in total. Area 58 was also identified as an area of increased clinic attendance of people without HIV (Fig B in S1 Text). Despite containing the most venues (n = 99), Area 25 did not have any AHI clusters.

The space-time permutation test captured only 11 (8%) participants with AHI, while covering 0.25% of the populated land area and 111 days (Table A in S1 Text), despite the α value of 0.5 that we chose to favor sensitivity in cluster detection. The two space-time permutation clusters mostly covered space also covered by the purely spatial clusters (Fig C in S1 Text). No spatiotemporal clusters of HIV-negative clinic attendees were observed.

### Phylogenetic analyses

There were 823 participants with HIV (652 with chronic HIV, 30 with recent HIV infection, and 141 with AHI). Due to sequencing capacity constraints, we prioritized people with AHI (n = 119 with blood samples), people with recent HIV (n = 29 with blood samples), and people with chronic HIV who referred or were referred by someone with AHI (n = 9). Additionally, we randomly chose and sequenced 20% (n = 80) of the remaining sample with chronic HIV who did not self-report being on ART, for a total of 89 with CHI. Of the 237 total samples that underwent sequencing, we obtained 222 sequences (112 AHI, 29 RHI, 81 CHI) for *gag* analyses and 222 (113 AHI, 29 RHI, 80 CHI) for *pol* analyses.

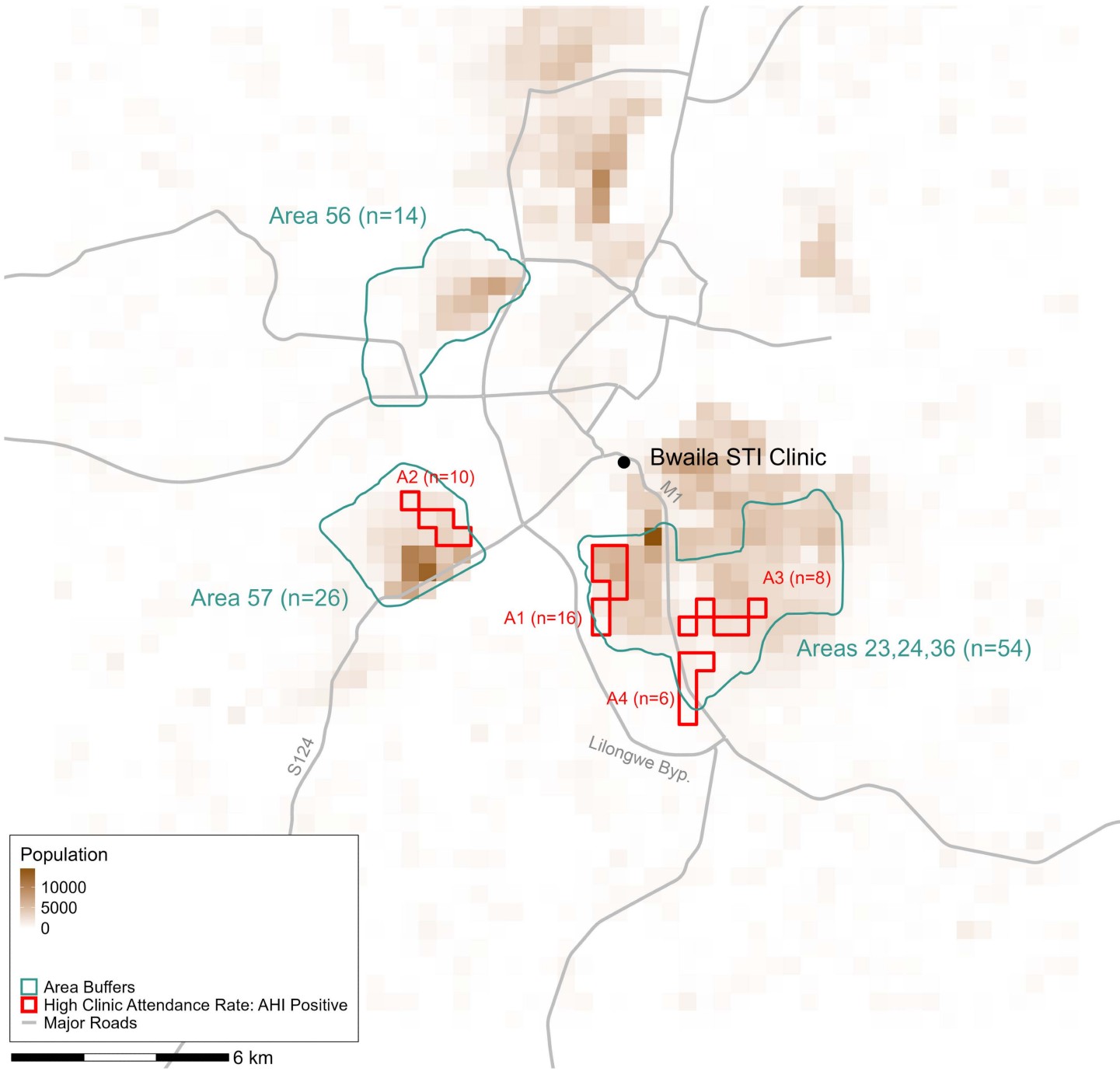

**Fig 1. Purely Spatial Clusters of STI Clinic Attendance with AHI, Overlaid on Buffered Boundaries for Traditional Authorities with High AHI Counts and Underlying Population Counts.** *The study area was divided into ~500x500 meter boxes and high-resolution population data were aggregated and used as a denominator. The numerator was the count of clinic attendees with AHI residing in each box. Clustering was determined using a flexible scan statistic (α = 0.1). Cluster labels (A1-4) correspond to those in* Table 1. *Areas buffers correspond to the actual area and a 250-meter buffer around the borders of the area. "N" corresponds to the number of residences of people with AHI in that area, out of 135 with spatial data. Road shapefiles were obtained from OpenStreetMap (openstreetmap.org) under the Open Database License (ODbL) (openstreetmap.org/copyright).*

**Table 1. Purely Spatial Clusters of STI Clinic Attendance with AHI and without an HIV Infection.**

| Cluster ID | Area (km²) | Clustered People | Population aged 15–64 | P-Value | Count With Genetic Data | Possible Phylogenetic Linkages | Venue Count^ |
|---|---|---|---|---|---|---|---|
| Acute 1 (A1) | 1.66 | 16 | 16,857 | <0.01 | 11/16 | 0 | 9 |
| A2 | 1.19 | 10 | 6,951 | <0.01 | 9/10 | 0 | 5 |
| A3 | 1.19 | 8 | 5,056 | 0.02 | 6/8 | 0 | 5 |
| A4 | 1.19 | 6 | 2,363 | 0.03 | 3/6 | 0 | 17 |
| Negative 1 (N1) | 1.66 | 29 | 16,334 | <0.01 | – | – | 8 |
| N2 | 0.71 | 15 | 5,252 | <0.01 | – | – | 3 |
| N3 | 2.13 | 20 | 18,028 | 0.02 | – | – | 21 |
| N4 | 0.95 | 5 | 591 | 0.03 | – | – | 4 |
| N5 | 1.66 | 15 | 11,758 | 0.04 | – | – | 10 |
| N6 | 1.19 | 7 | 2,198 | 0.05 | – | – | 34 |

^Venue count includes only those with point locations.

Sequences were all subtype C, except for one subtype-C-like, one recombinant of C and A, one recombinant of C and G, and one subtype A. The evolutionary model with the lowest Bayesian information criterion was the general time reversible (GTR) model, with empirical base frequencies and allowing for invariable sites and the use of a free rate model, for both *gag* and *pol* (Table B in S1 Text).

All identified phylogenetic clusters were of size two. Ten male-female phylogenetic pairs were identified (Table 2), with 9/10 being clinic-referred sexual partners (capturing all nine clinic-referral pairs where both had genetic data). Of the ten linked male-female pairs, one was acute-recent, seven were acute-chronic, and two were chronic-chronic. Referred partners attended the clinic within 23 days of each other, while members of the non-referral pair attended 42 days apart. We also identified four same-sex pairs (three male-male, one female-female) meeting our genetic distance thresholds who did not refer each other to the clinic, potentially indicating a missing intermediary or same-sex transmission. The female-female pair attended the clinic 1017 days apart and members of each male pair attended within 100 days of each other. Two of the male-male pairs were acute-acute and lived between 2 and 5 km from each other. The third male-male pair was chronic-chronic and the female-female pair was acute-recent. Acute and recent nodes were distributed throughout the phylogenetic trees, with no clear structure for *gag* or *pol* (Figs 3, 4).

## Phylogeographic analyses

Within geospatial clusters, 29/40 (73%) of samples were successfully sequenced and included in *gag* and *pol* analyses. There were zero phylogenetic clusters within the geospatial clusters. Within spatiotemporal clusters, 8/11 (73%) samples were successfully sequenced and included in *gag* and *pol* analyses. There were zero phylogenetic clusters within the spatiotemporal clusters.

## Discussion

Clustering of acute and recent HIV infections within geographic areas and sexual networks, if present and identifiable, could support spatially tailored and/or network-based interventions to prevent onward transmission. In this analysis, we used spatial, phylogenetic, and phylogeographic methods to evaluate the presence of such clusters among STI clinic patients and referred members of their sociosexual networks in Lilongwe, Malawi. Our findings have implications for the design of interventions that radiate from AHI detection programs in STI clinics. This type of setting has been an efficient location for AHI detection in Lilongwe for more than two decades, with numerous studies demonstrating the feasibility of

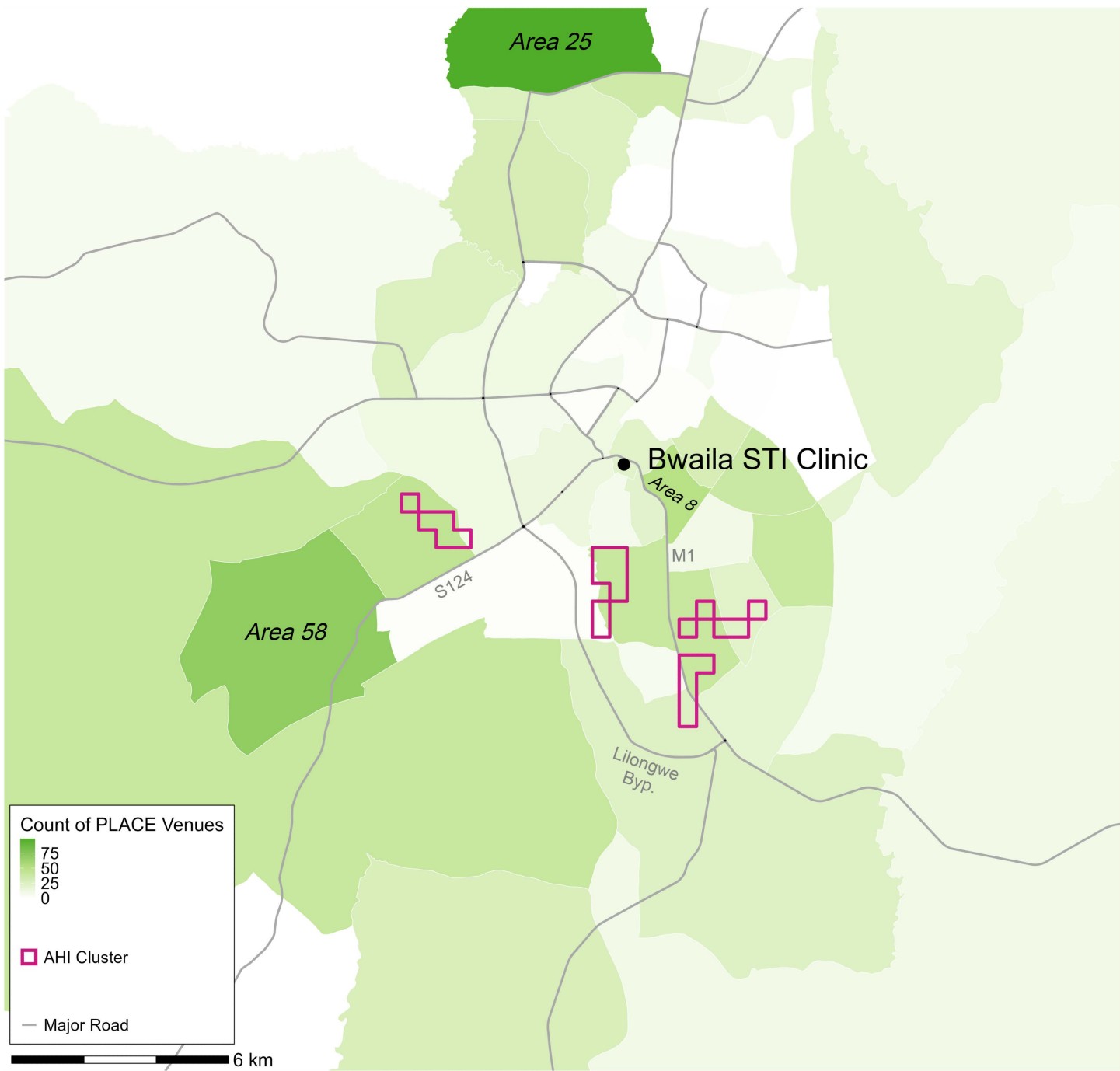

**Fig 2. Count of Venues Where People Meet New Sexual Partners at the Traditional Authority Level in Lilongwe, Overlaid with Major Roads and AHI Geospatial Clusters.** *Venues were identified using the PLACE method [31]. AHI geospatial clusters were identified using a flexible scan statistic and then a centroid was taken for cartographic purposes. White lines are traditional authority borders. Road shapefiles were obtained from OpenStreetMap (openstreetmap.org) under the Open Database License (ODbL) (openstreetmap.org/copyright).*

**Table 2. Characteristics of Phylogenetically Clustered Male-Female Pairs.**

| ID* | Sex (Age) | Clinic Referral Link? | Genetic Distance (*gag*) | Genetic Distance (*pol*) | Genetic distance (*whole*) | Temporal Distance |
|------|-----------|----------------------|--------------------------|--------------------------|----------------------------|-------------------|
| AR1 | M(23)-F(22) | no | 0.0000 | 0.0000 | 0.0002 | 42 days |
| AC1 | M(27)-F(21) | yes | 0.0027 | 0.0056 | 0.0091 | 0 days |
| AC2 | M(33)-F(21) | yes | 0.0033 | 0.0050 | 0.0110 | 10 days |
| AC3 | M(30)-F(21) | yes | 0.0034 | 0.0040 | 0.0058 | 23 days |
| AC4 | M(27)-F(22) | yes | 0.0041 | 0.0037 | 0.0104 | 8 days |
| AC5 | M(23)-F(22) | yes | 0.0046 | 0.0076 | 0.0038 | 6 days |
| AC6 | M(41)-F(38) | yes | 0.0075 | 0.0057 | 0.0074 | 0 days |
| CC1 | M(28)-F(26) | yes | 0.0109 | 0.0037 | 0.0138 | 13 days |
| AC7 | M(32)-F(32) | yes | 0.0206 | 0.0138 | 0.0300 | 0 days |
| CC2 | M(38)-F(35) | yes | 0.0267 | 0.0131 | 0.0320 | 14 days |

*A=Acute, R=Recent, C=Chronic, AR1 = Acute-Recent Pair 1.

such AHI detection programs and the potential promise of leveraging sociosexual networks to extend HIV prevention interventions beyond the clinic population [6,25,27,41–45].

In our spatial analysis, 30% of clinic attendees with AHI lived in 0.8% of the populated clinic catchment area, containing 3.5% of the population aged 15–64. These areas were typically close to or within dense population centers and connected by major roads to each other and areas with many venues where people meet new sexual partners. Interestingly, geospatial clusters of increased clinic attendance with AHI often overlapped or were contiguous with regions showing increased clinic attendance of people without HIV, which may be due to shared risk factors for both HIV and STI acquisition [46], as well as care-seeking dynamics driven by STI symptoms and/or perceived HIV/STI risk. The presence of two additional clusters of clinic attendees without HIV in or near two of the regions with the most venues is consistent with the care-seeking and shared-risk-factors explanation. The substantial overlap between AHI and HIV-negative clusters also suggests that areas with disproportionate STI clinic attendance, regardless of HIV status, could serve as efficient targets for both general and AHI-focused interventions, as they may represent areas at risk of high HIV/STI incidence. Areas of non-overlap (HIV-negative but no AHI clustering) could indicate potential inefficiencies of deploying AHI-specific interventions based on overall STI clinic attendance, or they may represent areas where we were not powered to detect AHI clusters or areas that may have the potential to be AHI clusters in the future.

To our knowledge, the geospatial distribution of AHI has not been previously described in any African setting. A 2022 study in Blantyre, Malawi attempted to identify acute and recent infections in clinics and venues across the city but found only one person with AHI and eight with recent infections [47]. A 2004–2014 study in KwaZulu-Natal, South Africa [14] identified three geospatial clusters of incident HIV infections, but the mean estimated time since infection was > 2 years in that study. Similar to our geospatial clusters along the M1 and S124, two of the KwaZulu-Natal clusters were adjacent to a national road. Road connectivity is an established facilitator of HIV transmission [48,49], and venues where people meet new sexual partners often cluster along major roadways [50]. In a 2019–2020 study of recent HIV infections (defined as <12 months after acquisition) in Malawi, three large-scale and three small-scale geospatial clusters of recent infections were identified [16]. However, the spatial analysis unit in that study was the health facility, complicating comparisons to our study and the study in KwaZulu-Natal [14], where the spatial analysis unit was the domicile. Despite methodological differences, each of these geospatial studies supports the existence of geospatial clustering of incident HIV infections, strengthening the case for spatially driven interventions for AHI detection.

Our phylogenetic analysis did not identify large outbreaks driven by transmission during early HIV infection among STI clinic attendees and their referred partners, nor any phylogenetic clusters containing more than two people. Previous

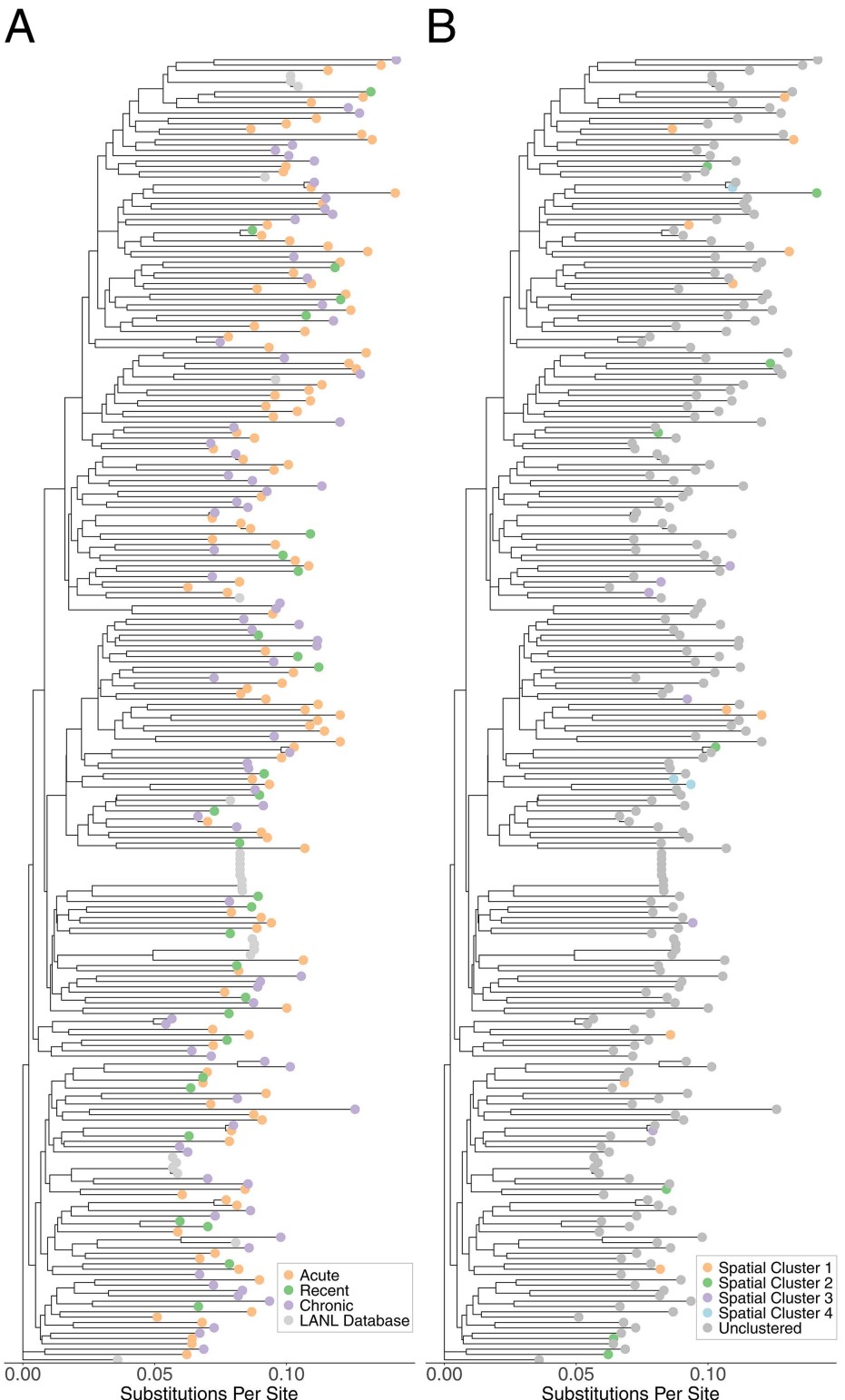

**Fig 3. Phylogenetic Tree (*gag*) of STI Clinic Attendees, colored by HIV stage (A) and Geospatial Cluster Membership (B).**

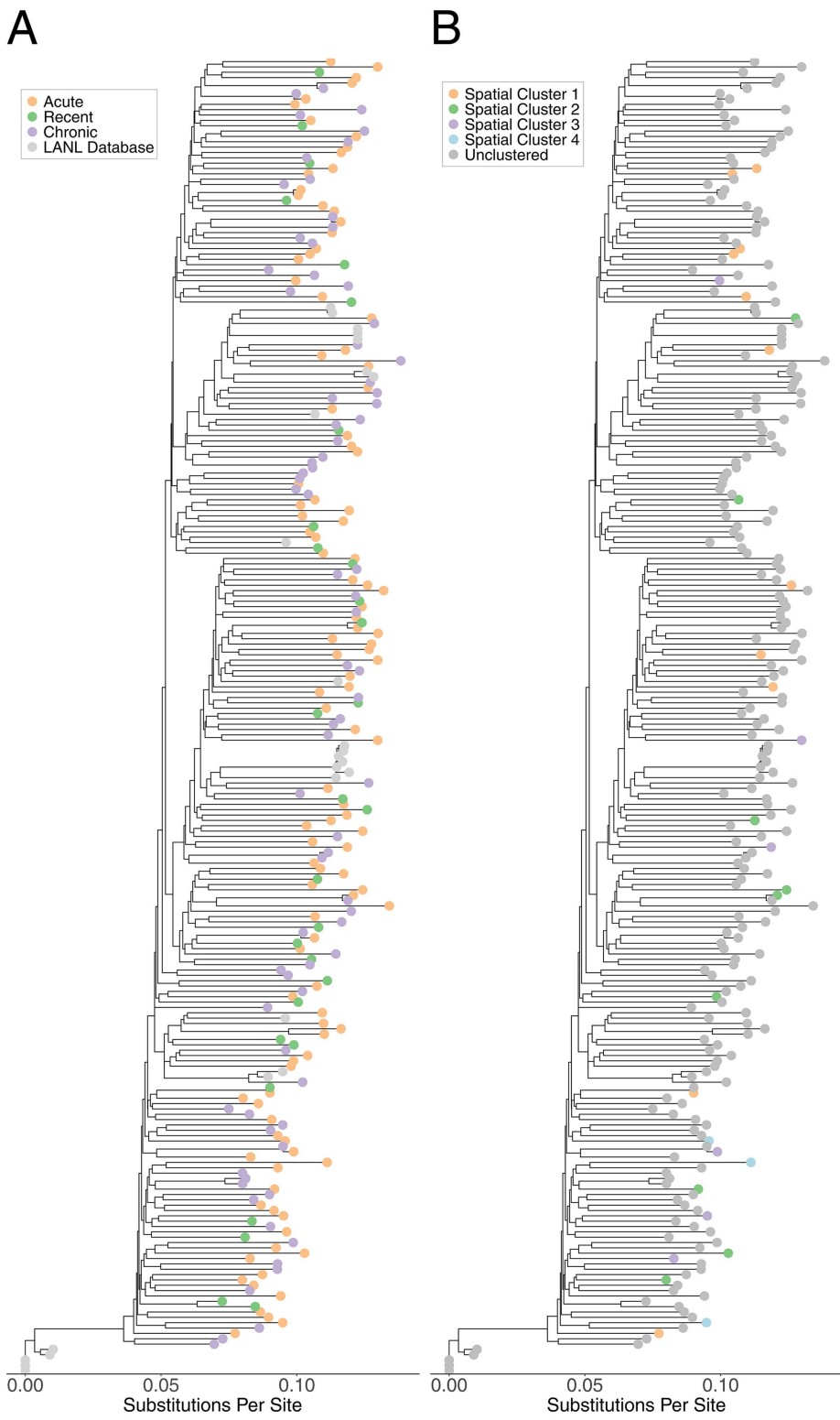

**Fig 4. Phylogenetic Tree (*pol*) of STI Clinic Attendees, colored by HIV stage (A) and Geospatial Cluster Membership (B).**

studies in Malawi [51] and other African settings [52,53] have predominantly identified transmission pairs rather than large chains, suggesting that long transmission chains caused by transmissions during early infection may be uncommon or difficult to identify in this region [54]. Furthermore, of the eight individuals with AHI phylogenetically linked to someone of the opposite sex, only one was linked to someone with AHI or RHI. Though these observations may appear to suggest a limited role for transmission during AHI, we note that our reliance on data from a single STI clinic population provides only a partial view of the full HIV transmission network in Lilongwe, and that our study was not designed to quantify the contribution of transmission during AHI to overall HIV incidence in Lilongwe. We also note that although we sequenced the majority of AHI/RHI participants and a randomly selected subset of CHI patients, our sampling design and sequencing priorities favored referred (vs. unreferred) sexual partners, and it is possible that links to unsequenced partners were missed or that transmissions occurring during AHI presented at the clinic as chronic-chronic pairs after the early stages had passed. We also identified two male-male acute-acute phylogenetic pairs who did not refer each other to the clinic. We could not determine whether these pairs represented same-sex transmission or a missing intermediary. It is possible that AHI may be of increased relevance or that clinic referral may be less effective in men who have sex with men compared to heterosexual populations in this setting.

Taken together, our spatial and phylogenetic results suggest that spatially focused AHI detection efforts could be efficient in identifying incident infections and prioritizing transmission prevention interventions, but may not facilitate the identification of rapidly developing, local transmission chains. The lack of phylogenetic links between members of geospatial clusters has at least three potential explanations. One explanation is that we insufficiently sampled the sexual networks in Lilongwe by only including STI clinic attendees and their referred partners; there were likely additional people with AHI, who were not referred to the clinic, who reside in those neighborhoods and they may have been part of a local transmission chain. Relatedly, transmission partners may have been unobserved due to high mobility or migration between regions outside the study area. Studies from Uganda, which find that migrants [55] and female partners of migrants [56] are at increased risk of HIV, support this possibility. A third possible explanation is that household locations may differ meaningfully from HIV transmission locations (e.g., bars, brothels, and other venues), and that transmission partners may reside in different areas of the city. The observed connectivity among AHI geospatial clusters, venues, and major roads is compatible with this possibility. We did not collect information on venue attendance from participants and thus could not conduct a more detailed investigation into the role of venues in HIV transmission, but our results suggest that future research examining links between AHI and venue attendance could be informative for intervention design.

As the largest existing study of AHI in sub-Saharan Africa and the first to use household GPS locations of people with AHI, our study highlights the potential of spatially tailored interventions for AHI detection. The extent to which rapidly propagating AHI transmission chains may be contributing to population-level HIV incidence is less clear from our study, as is the feasibility of identifying such chains through spatially focused and/or network-based interventions. Overall, our findings suggest that AHI detection programs anchored in STI clinic populations, their neighborhoods, and their sexual networks may be able to reach relatively large numbers of people with early HIV infection in efficient ways, enabling intervention during this most infectious period. The identification of these neighborhoods and networks may also enable the distribution of pre-exposure prophylaxis, HIV self-testing interventions, and combination prevention packages to those at high risk of infection or onward transmission. Future studies comparing the feasibility, reach, and population impacts of different tailoring methods can inform the optimal design and implementation of AHI-focused strategies to reduce HIV incidence.

## Supporting information

**S1 Text.**
(DOCX)

## Author contributions

**Conceptualization:** Griffin J Bell, Kimberly A. Powers, Mitch Matoga, Myron S. Cohen, Irving F. Hoffman, William C. Miller.

**Data curation:** Griffin J Bell, Pearson Mmodzi, Mitch Matoga, Edward Jere, Jane S. Chen, Courtney N. Maierhofer, Naomi Bonongwe, Esther Mathiya, Beatrice Ndalama, David Bonsall, Sharon S. Weir.

**Formal analysis:** Griffin J Bell, Jane S. Chen, Courtney N. Maierhofer, David Bonsall.

**Funding acquisition:** Griffin J Bell, Myron S. Cohen, Irving F. Hoffman, William C. Miller.

**Investigation:** Griffin J Bell, Kimberly A. Powers, Mitch Matoga, Myron S. Cohen, Irving F. Hoffman, William C. Miller.

**Methodology:** Griffin J Bell, Kimberly A. Powers, Oliver Ratmann, Ann M. Dennis, Maganizo B. Chagomerana, Sharon S. Weir, William C. Miller.

**Project administration:** Kimberly A. Powers, William C. Miller.

**Resources:** Kimberly A. Powers, Pearson Mmodzi, Edward Jere, David Bonsall, Sharon S. Weir, Mina C. Hosseinipour.

**Supervision:** Kimberly A. Powers, Oliver Ratmann, Ann M. Dennis, Sharon S. Weir, Mina C. Hosseinipour, Michael Emch, Myron S. Cohen, Irving F. Hoffman, William C. Miller.

**Validation:** Mitch Matoga, Edward Jere, Jane S. Chen, Courtney N. Maierhofer.

**Visualization:** Griffin J Bell.

**Writing – original draft:** Griffin J Bell, Kimberly A. Powers.

**Writing – review & editing:** Griffin J Bell, Kimberly A. Powers, Oliver Ratmann, Ann M. Dennis, Pearson Mmodzi, Mitch Matoga, Edward Jere, Jane S. Chen, Courtney N. Maierhofer, Sarah E. Rutstein, Kathryn E. Lancaster, Maganizo B. Chagomerana, Sharon S. Weir, Mina C. Hosseinipour, Michael Emch, Myron S. Cohen, Irving F. Hoffman, William C. Miller.

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
