## [Decision Letter · Decision Letter 0]

4 Sep 2025

PGPH-D-25-01598

Geospatial and Phylogenetic Clustering of Acute and Recent HIV Infections in Lilongwe, Malawi

Dear Dr. Bell,

Thank you for submitting your manuscript to PLOS Global Public Health. After careful consideration, we feel that it has merit but does not fully meet PLOS Global Public Health’s publication criteria as it currently stands. Therefore, we invite you to submit a revised version of the manuscript that addresses the points raised during the review process.

We look forward to receiving your revised manuscript.

Kind regards,

Sanghyuk S Shin

Academic Editor

Journal Requirements:

1. Please clarify all sources of funding (financial or material support) for your study. List the grants (with grant number) or organizations (with url) that supported your study, including funding received from your institution. 

2. State the initials, alongside each funding source, of each author to receive each grant.

3. State what role the funders took in the study. If the funders had no role in your study, please state: “The funders had no role in study design, data collection and analysis, decision to publish, or preparation of the manuscript.”

4. If any authors received a salary from any of your funders, please state which authors and which funders.

2. We ask that a manuscript source file is provided at Revision. Please upload your manuscript file as a .doc, .docx, .rtf or .tex.

3. We do not publish any copyright or trademark symbols that usually accompany proprietary names, eg (R), (C), or TM  (e.g. next to drug or reagent names). Please remove all instances of trademark/copyright symbols throughout the text, including ™ on page 5.

4. Please provide separate figure files in .tif or .eps format.

5. Some material included in your submission may be copyrighted. According to PLOS’s copyright policy, authors who use figures or other material (e.g., graphics, clipart, maps) from another author or copyright holder must demonstrate or obtain permission to publish this material under the Creative Commons Attribution 4.0 International (CC BY 4.0) License used by PLOS journals. Please closely review the details of PLOS’s copyright requirements here: PLOS Licenses and Copyright. If you need to request permissions from a copyright holder, you may use PLOS's Copyright Content Permission form.

Potential Copyright Issues:

Figure 1, 2, S1, S2: please (a) provide a direct link to the base layer of the map (i.e., the country or region border shape) and ensure this is also included in the figure legend; and (b) provide a link to the terms of use / license information for the base layer image or shapefile. We cannot publish proprietary or copyrighted maps (e.g. Google Maps, Mapquest) and the terms of use for your map base layer must be compatible with our CC-BY 4.0 license. 

6. We notice that your supplementary figures and tables are included in the manuscript file. Please remove them and upload them with the file type 'Supporting Information'. Please ensure that each Supporting Information file has a legend listed in the manuscript after the references list.

Reviewers' comments:

Reviewer's Responses to Questions

**Comments to the Author**

1. Does this manuscript meet PLOS Global Public Health’s publication criteria?

Reviewer #1: Yes

Reviewer #2: Yes

2. Has the statistical analysis been performed appropriately and rigorously?

Reviewer #1: Yes

Reviewer #2: No

3. Have the authors made all data underlying the findings in their manuscript fully available (please refer to the Data Availability Statement at the start of the manuscript PDF file)?

Reviewer #1: No

Reviewer #2: Yes

4. Is the manuscript presented in an intelligible fashion and written in standard English?

Reviewer #1: Yes

Reviewer #2: Yes

Reviewer #1: The authors present a well-written summary of largely null findings. I feel that the methodology is appropriate and the conclusions are well supported by the data. I have a few minor comments.

1. The major limitation of this study is the low sequencing coverage. The authors do sequence ~80% of participants with AHI, but only 25% of enrolled participants. The authors mention that they may have insufficiently sampled the sexual network by only focusing on STI clinic attendees; they should also discuss how the low sequencing coverage may impact their results.

2. The authors describe venues (bars, brothels, etc.) and the proximity of participant home locations to areas with these types of venues. It's not clear to the reader how we should interpret this. Is there any data (other than proximity) linking participants to these venues? For example, did any index cases report working at or visiting these specific areas?

3. The authors use genetic distance to infer likely recent transmission. This is not inappropriate, but I wonder if they considered model-based approaches to infer transmission events (perhaps based on the timing of diagnosis and the pattern of site mutations)? Such an approach might reveal the presence and number of unobserved intermediates between more distantly related pairs.

4. The authors assert in the discussion (4th paragraph) that individuals may be less likely to refer very recent sexual partners. Why do the authors think that? My expectation is that people would be more likely to refer very recent partners (recency bias).

5. Circular phylogenies are hard to read; it is difficult to compare branch lengths when they aren't plotted on the same horizontal axis. Please change these to rectangular trees and include a scale bar (subs/site) to improve interpretation.

Reviewer #2: Title: Geospatial and Phylogenetic Clustering of Acute and Recent HIV Infections in Lilongwe, Malawi

Overall Assesment

This manuscript tackles a highly relevant public health challenge detecting and characterizing clustering of acute and recent HIV infections (AHI/RHI) in Malawi using a combination of geospatial statistics and phylogenetic analysis. The dual approach is innovative and contributes significantly to understanding early HIV transmission dynamics. The findings have implications for designing targeted HIV prevention strategies, especially in urban African settings where generalized epidemics coexist with concentrated sub-epidemics.

The manuscript is well-motivated and methodologically rich, but there are areas where clarity, integration of findings, and framing of public health implications can be strengthened.

Major concerns

1. 2019 dataset is quite old, a more recent one would be more informative given the evolving nature of the epidemic.

2. While the spatial and the phylogenetic analysis are rigorous, they appear to operate in parallel. The key result is no phylogenetic clusters overlapped with geospatial clusters. This is important but under-discussed. The discussion should explore why this might be the case for example high partner mobility, clinic referral bias, or the possibility that transmission occurs in social/sexual venues rather than within neighborhoods.

3. Methodology needs to be expounded, even if it is under supplemetary materials: Evolutionary models; what are they, how many were they, what was the BIC values for the models, what is IQTree and so on.

4. AHI vs. RHI definitions are clear, but the distinction between acute, recent, and chronic infections could be summarized more succinctly in the results/discussion to aid readability for non-specialist readers.

5. The choice of 500×500m grids is justified, but results from the 1×1 km sensitivity analysis should be more explicitly reported in the results. Were clusters robust to this change?

6. The use of <2% distance for gag/pol and <4.5% for whole genome is appropriate, but the rationale for mixing these thresholds could be clarified. Why not apply a single genome-wide criterion?

7. The decision to allow looser clustering when gag/pol overlap <50% needs strongerjustification.

8. The authors conclude that AHI detection at STI clinics could guide interventions. This is valid, but could be strengthened by linking to life-course approaches and combination prevention strategies (e.g., community testing, PrEP, contact tracing).

9. More explicit discussion on feasibility in Malawi is warranted: would STI clinics realistically serve as sentinel sites given health system capacity constraints?

Minor concerns

1. The abstract states “three phylogenetic pairs consisted of only acute/recent members” percentages or context (e.g., proportion of all clusters) would help.

2. Figure/table clarity: ensure spatial maps are legible with population density shading and case locations.

3. Avoid overuse of acronyms (AHI, RHI, CHI, PLACE, etc.) define clearly at first use in abstract and main text.

**Do you want your identity to be public for this peer review?** For information about this choice, including consent withdrawal, please see our Privacy Policy

Reviewer #1: No

Reviewer #2: **Yes: ** Elphas Luchemo Okango

---

## [Editor Report · Decision Letter 1]

20 Oct 2025

Geospatial and Phylogenetic Clustering of Acute and Recent HIV Infections in Lilongwe, Malawi

PGPH-D-25-01598R1

Dear Mr. Bell,

We are pleased to inform you that your manuscript 'Geospatial and Phylogenetic Clustering of Acute and Recent HIV Infections in Lilongwe, Malawi' has been provisionally accepted for publication in PLOS Global Public Health.

Best regards,

Sanghyuk S Shin

Academic Editor